# The CaMKII/MLC1 Axis Confers Ca^2+^-Dependence to Volume-Regulated Anion Channels (VRAC) in Astrocytes

**DOI:** 10.3390/cells11172656

**Published:** 2022-08-26

**Authors:** Maria Stefania Brignone, Angela Lanciotti, Antonio Michelucci, Cinzia Mallozzi, Serena Camerini, Luigi Catacuzzeno, Luigi Sforna, Martino Caramia, Maria Cristina D’Adamo, Marina Ceccarini, Paola Molinari, Pompeo Macioce, Gianfranco Macchia, Tamara Corinna Petrucci, Mauro Pessia, Sergio Visentin, Elena Ambrosini

**Affiliations:** 1Department of Neuroscience, Istituto Superiore di Sanità, 00169 Rome, Italy; 2Department of Chemistry, Biology and Biotechnology, University of Perugia, 06123 Perugia, Italy; 3Core Facilities (FAST), Istituto Superiore di Sanità, 00169 Rome, Italy; 4Department of Medicine and Surgery, LUM Giuseppe Degennaro University, 70010 Bari, Italy; 5National Centre for Rare Diseases, Istituto Superiore di Sanità, 00169 Rome, Italy; 6National Centre for Drug Research and Evaluation (FARVA), Istituto Superiore di Sanità, 00169 Rome, Italy; 7Department of Physiology and Biochemistry, Faculty of Medicine and Surgery, University of Malta, MSD2080 Msida, Malta; 8Department of Physiology, College of Medicine and Health Sciences, United Arab Emirates University, Al Ain P.O. Box 17666, United Arab Emirates

**Keywords:** brain edema, astrocytes, Ca^2+^ release, ICl,_swell_, VRAC, CaMKII, leukodystrophy, rare diseases

## Abstract

Astrocytes, the main glial cells of the central nervous system, play a key role in brain volume control due to their intimate contacts with cerebral blood vessels and the expression of a distinctive equipment of proteins involved in solute/water transport. Among these is MLC1, a protein highly expressed in perivascular astrocytes and whose mutations cause megalencephalic leukoencephalopathy with subcortical cysts (MLC), an incurable leukodystrophy characterized by macrocephaly, chronic brain edema, cysts, myelin vacuolation, and astrocyte swelling. Although, in astrocytes, MLC1 mutations are known to affect the swelling-activated chloride currents (ICl,_swell_) mediated by the volume-regulated anion channel (VRAC), and the regulatory volume decrease, MLC1′s proper function is still unknown. By combining molecular, biochemical, proteomic, electrophysiological, and imaging techniques, we here show that MLC1 is a Ca^2+^/Calmodulin-dependent protein kinase II (CaMKII) target protein, whose phosphorylation, occurring in response to intracellular Ca^2+^ release, potentiates VRAC-mediated ICl,_swell_. Overall, these findings reveal that MLC1 is a Ca^2+^-regulated protein, linking volume regulation to Ca^2+^ signaling in astrocytes. This knowledge provides new insight into the MLC1 protein function and into the mechanisms controlling ion/water exchanges in the brain, which may help identify possible molecular targets for the treatment of MLC and other pathological conditions caused by astrocyte swelling and brain edema.

## 1. Introduction

In the central nervous system (CNS), ion and water movements are strictly controlled by astrocytes, the glial cell population mainly responsible for the maintenance of tissue homeostasis and ensuring correct brain functionality [1,2,3]. Alterations of these regulative processes can lead to serious complications, due to brain water accumulation and edema formation, as observed in several CNS injuries such as ischemia, trauma, infection, tumors, and inflammation [3,4,5]. The astrocyte’s favorable morphological structure and anatomical distribution, with end-foot processes contacting blood vessels, allow these cells to connect parenchymal tissue to the blood circulation, thus favoring the uptake and removal of accumulating solutes and water [2,6]. To accomplish this task, astrocytes also express a large number of transporters and ion channel proteins [1,5,6,7]. Understanding how these molecules interplay would greatly help identify potential targets to treat the devastating consequences of brain edema [8]. Interesting advances in the comprehension of these mechanisms emerged from the study of genetic diseases distinguished by chronic brain edema and caused by mutations in specific astrocytic proteins, such as MLC1 in the megalencephalic leukoencephalopathy with subcortical cysts (MLC) disease [9]. This rare leukodystrophy, which causes severe motor and cognitive impairments and epilepsy [10,11], is characterized by diffuse white matter edema and subcortical cysts, observed by magnetic resonance imaging, and by myelin vacuolation and astrocyte swelling, as revealed by histological analysis of MLC brain tissue. In 80% of MLC patients, the disease is linked to mutations in the *MLC1* gene encoding the MLC1 protein [12], a 377 amino acid protein containing eight predicted transmembrane domains and short cytoplasmic amino and carboxylic tails. In the human and mouse brain MLC1 is almost exclusively expressed by astrocyte end-feet contacting blood vessels and meninges, as well as by the Bergmann glia of the cerebellum [13,14]. Although, to date, MLC1′s proper function has remained elusive, both cellular and animal models of MLC have proven that a lack of functional MLC1 affects the activation of two different chloride currents: an outward-rectifying anion current, mainly due to volume-regulated anion channels (VRAC); and a strong inward-rectifying chloride current mediated by the voltage-gated ClC-2 channels [15,16,17,18]. Both currents are affected by knock-out (KO) or mutations of the *MLC1* gene, but they may differently contribute to MLC pathogenesis. Indeed, while the ClC-2-activated current is likely involved in extracellular potassium ion (K^+^) homeostasis during neuronal activation [17], VRAC, the main component of the swelling-activated Cl^-^current (ICl,_swell_), is mainly responsible for regulatory volume decrease (RVD) in response to osmotic changes [18]. Interestingly, MLC1 was described to interact with a number of effector proteins directly involved in osmotic homeostasis, such as AQP4, Na, K-ATPase, TRPV4, and Kir4.1 [19,20,21,22]. All these experimental and clinical observations suggest that MLC1 mutation-induced ion/water exchange defects are responsible for homeostatic disorders such as edema or myelin vacuolation, but how this protein regulates cell volume and the underlying anion currents is unknown. To clarify this issue, we reasoned about the possible connection between MLC1 and intracellular Ca^2+^ signaling. This explorative working hypothesis stems from our previous findings indicating MLC1 expression in caveolar rafts [21,22,23], plasma membrane (PM) microdomains hosting several proteins involved in Ca^2+^ signaling [24], and MLC1′s potentiation of Ca^2+^ influx through TRPV4 channels in response to osmotic stress [20]. In addition, defects in intracellular Ca^2+^ homeostasis were observed in different MLC pathological models [25,26]. Notably, in astrocytes, Ca^2+^ regulates all MLC1-modulated biological processes, such as ICl_,swell_ and RVD [27]. By integrating complementary experimental approaches, encompassing biochemical, molecular biology, imaging, proteomic, and electrophysiological techniques being applied to astrocyte-based cellular models of MLC, we revealed new Ca^2+^-dependent functional properties of the MLC1 protein that can unravel the molecular mechanisms controlling volume changes in astrocytes and whose dysfunction accounts for brain edema in MLC patients.

## 2. Materials and Methods

### 2.1. Cell Cultures and Treatments

Astrocyte-enriched cultures were generated from newborn CD1 Swiss mice and maintained in culture as previously described [28]. Generation/maintenance of U251 cell lines expressing wild type (MLC1-WT) or mutated MLC1 (MLC1-S280L) has been previously described [20]. Cell treatments were performed in serum-free (SF) medium, as follows: 5 min with adenosine triphosphate (ATP, 100 µM, Sigma-Aldrich, Saint Louis, MO, USA), 30 min with isotonic solution (140 mM NaCl, 5 mM KCl, 2 mM MgCl_2_, 2 mM CaCl_2_, 20 mM D-glucose, 5 mM HEPES, pH 7.4) or hypotonic solution prepared by adding 30% distilled water to the isotonic solution (Hypo) [29]. For high K^+^ stimulation, cells were treated for 30 min with the same isotonic buffer, with the KCl concentration increased up to 60 mM (HK), as previously described [30]. For CaMKII inhibition U251 MLC1-WT cells and mouse astrocytes were treated for 15 min with 10 µM of KN93 (10 µM, Sigma-Aldrich, Saint Louis, MO, USA) in isotonic solution or co-stimulated for 15 min with 10 µM KN93 in hypotonic solution. Cycloheximide (CHX) treatment (100 µg/mL, Sigma-Aldrich) was maintained for 1, 3, 4, or 5 h.

### 2.2. Generation of U251 Cell Lines Expressing MLC1-T17A and T17D Mutants

Mutagenesis of the amino acid (aa) Threonine (T) into Alanine (A) or aspartic acid (D) (T17A and T17D, respectively) was obtained by using oligonucleotides containing the mutations required. pQCXIN-MLC1-WT plasmid construct [20] was digested with MfeI and NotI restriction enzymes (NEB, New England Biolabs Ltd., Hitchin, UK). The DNA fragment obtained was run on agarose gel and extracted/purified with a Gel Extraction Kit (Qiagen, Hilden, Germany). gBlocks™ Gene oligonucleotides (IDT, Integrated DNA Technologies, Inc., Coralville, IA, USA) containing a T17A or T17D mutation and MfeI/NotI sequence at their ends (18 ng for each fragment) were incubated for 1 h at 50 °C with 100 ng of the MfeI/NotI digested plasmid in 1X Gibson Assembly^®^ Master Mix (NEB, New England Biolabs). Then, 2.0 µL of the reaction mixture were used to transform DH5α competent cells. Positive colonies grown on the selective agar medium were screened by plasmid DNA sequencing (Eurofins, Genomics, Ebersberg, Germany). Plasmids containing the desired mutations were used for stable U251 line generation, as previously described [20].

### 2.3. Immunofluorescence Staining (IF)

IF staining was performed as described in [26,30], using the following primary antibodies (Abs): anti-MLC1 polyclonal Ab (pAb) (1:50, Atlas AB, AlbaNova University Center, Stockholm, Sweden) and anti-Xpress monoclonal Ab (mAb) (1:50, ThermoFischer Scientific, Rockford, IL, USA) to detect the Xpress tag present at the NH_2_ terminal of the recombinant MLC1 and anti-phospho (p)CaMKII pAb (1:50, Santa Cruz Biotechnology, Dallas, TX, USA). As secondary Abs, a biotinyated goat anti-rabbit IgG H+L Ab (4.3 μg/mL; Jackson Immunoresearch Laboratories, West Grove, PA, USA), followed by streptavidin-TRITC (2 μg/mL; Jackson) and Alexa Fluor 488 goat anti-mouse IgG Ab (1:300, Invitrogen, Thermo Fisher Scientific, Rockford, IL, USA) were used. Coverslips were sealed in Fluoroshield with DAPI (Sigma-Aldrich) and analyzed with a fluorescence microscope (Carl Zeiss, Jena, Germany, ZEISS Axiocam 512); images were acquired using ZEN 3.1 blu edition software. The distribution of immunofluorescence pixel intensity along a freely defined line (indicated by a dotted arrow in the IF images) spanning the whole cell was evaluated using the profile analysis tool of the NIH ImageJ software in 50–60 cells/condition, as described in [31]. The colocalization signal was evaluated by Pearson’s correlation coefficient (PCC). Statistics analysis was performed using one way ANOVA and an unpaired two-tailed Student’s *t*-test.

### 2.4. Pull-Down Assay and Immunoprecipitation

Pull-down assay was performed using rat brain extracts and sepharose-immobilized GST-fused MLC1 (C-t and N-t domains) and with His-fused MLC1 full-length protein (FL), all obtained as previously described [20]. Rat brain was extracted with ice-cold binding buffer (25 mM Tris-HCl, pH 7.4, 150 mM NaCl, 1.5 mM MgCl_2_, 1% Triton X-100, 0.5% sodium deoxycolate), containing a protease inhibitor mixture (Roche Molecular Biochemicals, Mannheim, Germany). Brain extracts (2 mg/mL) were incubated with the indicated fusion proteins (20 µL of beads suspension) followed by centrifugation and WB analysis. For immunoprecipitation of U251 cells and primary mouse astrocytes, cells were lysed in ice-cold RIPA buffer. The supernatants were incubated with 50% (*w*/*v*) protein G plus agarose beads (Santa Cruz Biotechnology) for 2 h at 4 °C and left overnight (ON) at 4 °C in a rotating wheel with the anti-Xpress mAb or the anti-pCaMKII pAb, respectively, previously cross-linked with dimethyl pimelimidatedihydrochloride (DMP, Sigma Aldrich). The immunoprecipitated eluates were processed for western blot analysis (WB).

### 2.5. Protein Extract Preparation and Western Blotting (WB)

Total protein extracts from astrocytoma cell lines, quantification, and WB analysis were performed as previously described [20,21]. Nitrocellulose membranes were blotted ON at 4 °C using anti-MLC1 pAb (1:1500, in-house generated [22]), anti-Actin mAb (1:2000), anti-pCaMKII pAb (1:200), anti-PP1 pAb (1:1000), anti-PP2A pAb (1:1000), anti-phosphothreonine mAb (1:500) (all from Santa Cruz Biotechnology), anti β-DG mAb (1:250; NCL-43 DAG, Novocastra Lab, Ltd., Newcastle upon Tyne, UK), and anti-Xpress mAb (1:2000, Thermo Fisher Scientific), in PBS + 3% BSA. Densitometric analyses of WB experiments were performed using NIH ImageJ software and Bio-Rad ChemiDoc XRS system.

### 2.6. In Vitro Kinase Assay for CaMKII

The phosphorylated active enzyme CaMKII was immunoprecipitated from U251 cells using a specific anti-pCaMKII (Thr286) pAb (2µg, Santa Cruz Biotechnology). The enzyme was then used to phosphorylate in vitro the agarose-bound-GST N-terminal peptide and the His-fused MLC1 FL protein obtained as described above. The in vitro kinase assay was performed by incubating the immunoprecipitated pCaMKII or the commercial recombinant purified pCaMKII (0.5 µg, Invitrogen, Thermo Fisher Scientific) with the recombinant proteins in kinase buffer (25 mM Hepes, pH 7.4, 50 mM KCl, 10 mM MgCl_2_, 1 mM DTT, 2 mM CaCl_2_, 0.1 mM PMSF, phosphatase inhibitor cocktail) containing 2 µM CaM and 1–2 μCi [γ^32^P] ATP (>3000 Ci/mmol) for 10 min at 30 °C under gentle stirring. When indicated, the CaMKII inhibitor KN93 (10 µM, Sigma-Aldrich) was used in the reaction mixture. The kinase reaction was stopped by adding 4X SDS loading buffer and the samples were resolved on 12% SDS-PAGE. The dried gels were exposed to X-ray film for autoradiography. When indicated, myelin basic protein (MBP) was added to the reaction mixture.

### 2.7. Mass Spectrometry Analysis

For MS analysis, the phosphorylation reaction was carried out in kinase buffer containing 2 µM CaM and 0.3 mM ATP, using commercial recombinant purified pCaMKII (see above). Proteins eluted from the beads were separated on a 1D-gel NuPAGE 4–12% (Novex, Invitrogen, Carlsbad, CA, USA). The bands were stained with Coomassie blue (Colloidal Blue Staining kit, Invitrogen), cut, and digested with trypsin (Promega, Madison, WI, USA), Asp-N-protease (Promega), or Glu-C endoproteinase (Sigma-Aldrich). To identify phosphorylated sites, the peptide mixture was analyzed using LS-MS/MS as described [32]: MS/MS and MS/MS/MS were acquired in order to search for phosphopeptides carrying mass increases and neutral losses, respectively, and all the data were analyzed using the SEQUEST algorithm [33]. Data are available via ProteomeXchange with identifier PXD030549. 

### 2.8. Electrophysiology

The whole cell dialyzed configuration, the external standard, and Ca^2+^-free (0 Ca^2+^) solutions, as well as the internal solution, were used for electrophysiological recordings of the ICl,_swell_, as previously described [34]. Briefly, the external standard solution contained: NaCl 140 mM, KCl 5 mM, CaCl_2_ 2 mM, MgCl_2_ 2 mM, MOPS 5 mM, and glucose 10 mM (pH 7.40). The external Ca^2+^-free solution contained NaCl 140 mM, KCl 5 mM, MgCl_2_ 4 mM, MOPS 5 mM, and glucose 10 mM (pH 7.40). The internal solution contained KCl 155 mM, EGTA-K 1 mM, MOPS 5 mM, and MgCl_2_ 1 mM (pH 7.20). The external hypotonic solution was obtained by adding 30% distilled water to the external solution. VRAC-mediated ICl,_swell_ were blocked by DCPIB (10 µM), while KN93 (10 µM, Sigma-Aldrich) was used for CaMKII inhibition. Both compounds were dissolved in DMSO (Sigma-Aldrich), with the highest DMSO concentration in the recording solutions being 0.1%. Both the degree and kinetic of inactivation of the hypotonic-induced ICl,_swell_ were assessed in current traces elicited at +100 mV. The degree of inactivation was obtained by dividing the current amplitude taken at the steady-state (I_steady-state_) and the current amplitude taken at the peak (instantaneous current or I_istantaneous_), as follows: 1-(I_steady-state_/I_istantaneous_). The kinetic of inactivation was calculated by fitting the current trace with a monoexponential decay function, which provided the time constant (τ) of current inactivation. To verify the involvement of any Ca^2+^ possibly released from intracellular stores, we assessed the maximal ICl,_swell_ activated using hypotonic solution with 0 mM Ca^2+^ (to avoid contamination from external Ca^2+^ influx) in U251 (MLC1-WT) cells pretreated or not with 1 µM thapsigargin (TG, Sigma-Aldrich), a SERCA pump irreversible inhibitor. Before the activation of the ICl,_swell_ with the Ca^2+^ free hypotonic solution, cells were incubated with 1 µM TG in an isotonic Ca^2+^-free solution for 5 min, a time period sufficient to completely deplete the intracellular Ca^2+^ stores in non-excitable cells. To verify the involvement of IP3 receptor (IP3R)-mediated Ca^2+^ release from internal stores, we assessed the maximal ICl,_swell_ activated by the re-addition of Ca^2+^ to the hypotonic solution in WT cells in the presence and absence of 1 mM 2-Aminoethoxydiphenyl borate (2-APB, Sigma-Aldrich), an IP3R inhibitor, added in the internal pipette solution. We used this protocol to reach a high concentration of 2-APB in the cell cytoplasm and to avoid the direct inhibition of VRAC occurring in U251 cells in response to the extracellular Ca^2+^ re-addition in the constant presence of 100 µM 2-APB, as previously described [35]. 

### 2.9. Fura-2-Based Ca^2+^ Imaging

Cytoplasmic Ca^2+^ was measured using the fluorescence video-imaging technique with the Ca^2+^ indicator Fura-2-AM (Invitrogen, Thermo Fisher Scientific), as previously described [20]. U251 cells seeded on poly-l-lysine treated glass coverslips were loaded with 2 μM Fura-2-AM dissolved in recording buffer, with the following composition (mM): 140 NaCl, 5 KCl, 1 MgCl_2_, 2.5 CaCl_2_, 5.5 D-glucose, and 10 HEPES/NaOH at RT (pH 7.4). In Ca^2+^-free solutions, Ca^2+^ was replaced with MgCl_2_ and 0.5 mM EGTA was added (0-Ca^2+^ solution). A custom-made local solution exchange system allowed the solution bathing the cells to be rapidly switched between control and test solutions. Ca^2+^ concentrations were expressed as the ratio of emissions at 340 nm to 380 nm.

### 2.10. Statistical Analysis

All the statistical analyses were performed using GraphPad prism software (Version 9.3, GraphPad, San Diego, CA 92108, USA). Results were expressed as mean ± standard error of mean (SEM). Data were first subjected to a normality test (D’Agostino and Pearson Omnibus Normality test); when data followed a normal distribution, a Student’s *t*-test was applied; otherwise, non-parametric tests, such as Wilcoxon test or Kruskal-Wallis test, followed by Dunn’s Multiple Comparison post hoc test, when necessary, were performed. Statistically significant *p* values are * *p* < 0.05, ** *p* < 0.01 and *** *p* < 0.001, while not significant differences are indicated with the “ns” abbreviation.

## 3. Results

### 3.1. Ca^2+^-Dependent Protein Kinases II (CaMKII) Binds MLC1 Protein in Astrocytes

To explore the link between MLC1 and Ca^2+^ signaling in astrocytes, we focused on the possible involvement of the Ca^2+^-dependent CaMKII enzyme. The recognized role of the Ca^2+^-dependent kinases (PKC and CaMKII) in astrocyte volume regulation [3] and the observation of the presence of a potential CaMKII phosphorylation target site (RXXS/T) in a highly conserved amino acid (aa) sequence of the MLC1 protein (Figure 1A) underpinned this choice. CaMKII is a multifunctional serine/threonine kinase regulated by the Ca^2+^/Calmodulin (CaM) complex and recognized as a crucial mediator of many physiological effects caused by elevations of intracellular Ca^2+^ [36]. To study the relationship between MLC1 and CaMKII, we used the already characterized U251 human astrocytoma cell lines stably overexpressing the human recombinant MLC1 wild-type (MLC1-WT), [20,23,26]. We first verified a possible interaction between the phosphorylated (p), activated CaMKII, and MLC1 protein in these cells, by performing immunofluorescence (IF) staining and co-immunoprecipitation (IP) experiments. These analyses showed that pCaMKII/MLC1 colocalization was poorly detectable in the cytosol of U251 cells in unstimulated conditions (Figure 1B), but it increased in perinuclear/ER areas after cell treatment with hyposmotic and high K^+^ solutions (Figure 1C–E, asterisks in panel C and arrowheads in panel D, respectively), with both conditions favoring intracellular Ca^2+^ increase and MLC1 functional activation [15,17]. Co-IP experiments in the same cells confirmed an MLC1/CaMKII interaction at the biochemical level, by showing that pCaMKII co-immunoprecipitated with MLC1 (Figure 1F, arrowhead and arrow, respectively). Accordingly, pCaMKII and MLC1 co-IP were observed when mouse astrocyte protein extracts were used, revealing that the endogenous MLC1 protein expressed in astrocytes from primary cultures also interacted with the pCaMKII (Appendix A).

To better clarify the molecular details of MLC1/CaMKII binding, we also performed a pull-down assay of rat brain protein extracts using recombinant GST-MLC1-NH_2_-t (GST-MLC1-N-t) and GST-MLC1-COOH-t (GST-MLC1-C-t) domains or the full length MLC1 recombinant protein (aa 2-377, FL) containing a Histidine (His) tag, which were immobilized, respectively, on glutathione-agarose or Histidine-binding resin, as previously described [19,22]. pCaMKII was observed among the pulled-down proteins when GST-MLC1-C-t and the FL protein were used, but not when the assay was performed with the recombinant GST-MLC1-N-t peptide (Figure 1G). The MLC1 GST-MLC1-C-t and FL protein also bound the serine/threonine protein phosphatase 1 (PP1) and protein phosphatase 2A (PP2A), (Figure 1G) with both enzymes involved in the de-phosphorylation of the CaMKII and of their target proteins [37,38]. In the same experiment, no binding occurred with beta dystroglycan (β-DG), a protein already found not to interact with either the MLC1 N- or C-terminal [19], and thus used as negative control.

### 3.2. CaMKII Phosphorylates MLC1 Protein at the Amino Acid Residue Threonine 17

The above results identified MLC1 as a CaMKII-binding protein. To verify whether MLC1 was also phosphorylated by CaMKII, we performed an in vitro kinase assay. The assay was carried out in the presence of [γ^32^P] ATP, using the full-length MLC1 protein (FL) and the GST-MLC1 NH_2_-t (where the putative CaMKII phosphorylation motif had been observed) as substrates for the CaMKII enzyme precipitated from U251 cells. As shown in Figure 2A, the immunoprecipitated phosphorylated enzyme (pCaMKII) was fully functional, being able to phosphorylate the myelin basic protein (MBP), a control substrate used to monitor CaMKII activity. The CaMKII enzyme also phosphorylated the FL-MLC1 protein (Figure 2B) and the MLC1-N-t peptide (Figure 2C). The specificity of this phosphorylation was confirmed by introducing to the reaction mixture the CaMKII inhibitor KN93, which strongly reduced MLC1 protein phosphorylation in both cases (Figure 2B,C). To identify the specific aa residue targeted by the CaMKII enzyme, a kinase assay was performed with the use of the CaMKII recombinant enzyme and the GST-MLC1 NH_2_-t peptide and the reaction mixture was subjected to LC-MS/MS analysis. These experiments allowed the recognition of a phosphorylation site at Threonine (T), in position 17 of the MLC1-N-t peptide, within the supposed CaMKII consensus sequence RXXS/T (aa 14–17, Figure 2D).

### 3.3. CaMKII-Mediated Phosphorylation of MLC1 in Response to Intracellular Ca^2+^ Increase Affects MLC1 Protein Assembly and Stability in U251 Cells

The activation of CaMKII and the consequent CaMKII-mediated phosphorylation can strongly modulate the target protein structural organization, intracellular localization, and functionality [38,39]. To clarify the effects of CaMKII-mediated phosphorylation on MLC1 structural and functional properties, we generated two U251 cell lines, where the CaMKII phosphorylation site T17 was mutated into (i) alanine (T17A), a non-phosphorylatable amino acid, or (ii) aspartic acid (T17D), an amino acid mimicking protein constitutive phosphorylation. Characterization of these cell lines by WB and IF revealed that T17D substitution favored dimer formation (Figure 3A, arrow) and a general increase of MLC1 protein distribution, either in the cytoplasm or at cell PM, when compared to MLC1-WT expressing cells (Figure 3B, arrowheads and panel C). No strong alterations of MLC1 intracellular distribution were observed in cells expressing the non-phosphorylatable MLC1-T17A mutant by IF staining (Figure 3B–C), suggesting that CaMKII phosphorylation did not affect MLC1 protein trafficking. In the U251 cell lines that we generated, MLC1 regulation at transcriptional level could not occur, since the MLC1 cDNA is cloned under the control of a constitutive promoter (the cytomegalovirus promoter, CMV). Therefore, we tested whether CaMKII phosphorylation could increase MLC1-T17D protein levels by affecting protein stability. To this aim, we treated cells for different durations with cycloheximide (CHX), an inhibitor of protein biosynthesis, and verified by WB the degradation kinetics of the MLC1-T17D and T17A mutants in comparison with the MLC1-WT protein. CHX treatment showed that T17D mutation improved MLC1 stability, this latter protein still being detectable between 3 and 4 h of CHX treatment, a time when the MLC1-WT was already degraded (Figure 3D). In parallel, we also observed that the MLC1-T17A mutant had a slightly reduced stability when compared to MLC1-WT (Appendix A). These combined analyses encompassing WB, IF, and CHX treatment provided evidence that CaMKII-dependent phosphorylation favors MLC1 dimer formation and protein stabilization. 

Consistent with these results, we also observed that in both U251 cells expressing the recombinant MLC1-WT and in primary mouse astrocytes expressing the endogenous MLC1 protein, cell treatment with the CaMKII inhibitor KN93 reduced the formation of the MLC1 dimer in response to hyposmotic stimulation (Appendix A).

To have a more direct confirmation of the CaMKII-mediated phosphorylation of MLC1 at the T17 aa residue in an in vivo system, we monitored whether phosphorylation occurred in the MLC1-WT protein, and not in the non-phosphorylatable mutant T17A, in response to stimuli inducing an increase in intracellular Ca^2+^ levels and CaMKII activation. To this aim, MLC1-WT and MLC1-T17A U251 cells were treated with ATP, one of the most efficient physiological activators of intracellular Ca^2+^ release in astrocytes and astrocytoma cells [40], and MLC1 protein was immunoprecipitated from control and stimulated cells. To monitor the presence of threonine (Thr) phosphorylation on MLC1 protein, we performed WB analysis of the immunoprecipitated protein eluates using a specific anti-phospho(p)Thr mAb, as previously described [41]. These experiments showed increased levels of p-Thr at the molecular weight expected for the MLC1 protein in the eluate derived from cells expressing the WT MLC1, but not from cells expressing the T17A mutant, in response to ATP stimulation (Figure 3E, arrow), thus providing evidence that CaMKII phosphorylation of MLC1 occurs at T17 in U251 cells and not only in in vitro systems.

Considering the assembled evidence of the above studies, we can conclude that, in astrocytes, MLC1 stabilization/dimerization is favored by the phosphorylation of the T17 aa residue induced by the CaMKII enzyme activated by intracellular Ca^2+^ release.

### 3.4. MLC1 Potentiates VRAC-Mediated ICl,_swell_ in U251 Cells 

As MLC1 was involved in the activation of Cl-currents (e.g., ICl,_swell_) and cell volume regulation in several MLC disease models [15,16,17], we investigated the role of CaMKII phosphorylation on MLC1 functionality by performing electrophysiological experiments on U251 cells. We first assessed whether in our U251-based cell model MLC1 was also involved in ICl,_swell_ potentiation. To this aim, patch-clamp recordings were performed on U251 cells expressing MLC1-WT and, as controls, on U251 cells transfected with an empty vector (Ø) or expressing the already characterized pathological mutant MLC1-S280L [20,26], (Figure 4). Step potentials from −100 to +100 mV were imposed before (left) and during (center) perfusion with hyposmotic solution; the current obtained by subtracting the current elicited under hypotonic (Hypo) solution and that in isotonic (Iso) solution represents the hyposmosis-induced current (ICl,_swell_, right), (Figure 4A). The quantitative analysis showed that the hyposmosis-induced ICl,_swell_ density was significantly larger in MLC1-WT than in Ø and MLC1-S280L cells (Figure 4B,C). Notably, the expression of MLC1-WT also affected the ICl,_swell_ inactivation (Figure 4 and Appendix A). Specifically, the inactivation kinetic, calculated as a time constant (τ), was significantly faster in MLC1-WT (112.1 ± 8.0 ms) compared to MLC1-Ø (154.3 ± 14.5 ms) cells, and the degree of inactivation, calculated as the ratio of the current amplitudes taken at the steady state and at the peak (1–(I_steady state_/I_peak_)), was significantly larger in MLC1-WT (0.86 ± 0.03) compared to MLC1-Ø (0.71 ± 0.02) cells. Overall, these findings further support the ability of MLC1 to also act as a ICl,_swell_ modulator when expressed in U251 cells. 

### 3.5. MLC1 Potentiates VRAC-Mediated ICl,_swelll_ in U251 Cells through a CaMKII-Dependent Mechanism

Once it was determined that the expression of functional MLC1 in U251 cells was required for the enhancement of the ICl,_swell_, we investigated if the CaMKII-dependent phosphorylation of MLC1 represented a key step for its activation and the modulation of ICl_,swell_ by MLC1. To this aim, we assessed the ICl,_swell_ density in U251 expressing MLC1-WT or carrying the T17A or T17D mutations. Voltage ramps from −100 to +100 mV from a holding potential of −40 mV were applied. In Figure 5A, we illustrate the protocol used and the time course of ICl_,swell_ activation in U251 cells with the hypotonic solution, together with representative current ramps (Inset: left) under hypotonic conditions and after application of 10 µM DCPIB, which selectively inhibits VRAC. The inset on the right of Figure 5A shows the typical ICl,_swell_ induced by step potentials during hyposmotic solution perfusion in the absence or presence of 10 µM DCPIB, which almost fully blocked the hyposmosis-activated currents. The results from these experiments revealed that the hyposmotic solution induced an inward current at −80 mV (ICl,_swell_) in all cell models, as shown in the bar plot of Figure 5B. As described above, MLC1-WT cells exhibited a greater current density (~54 pA/pF) than cells transfected with the empty vector (MLC1-Ø, not containing MLC1) (~32 pA/pF). By analyzing ICl,_swell_ from T17D and T17A mutants we observed that the T17D cells, exhibited an ICl,_swell_ density similar to WT cells (~56 pA/pF), while the ICl,_swell_ density observed in T17A cells (~32 pA/pF) was similar to MLC1-Ø cells (Figure 5B). These data suggest that CaMKII phosphorylation of MLC1 is required for upregulating ICl,_swell_. In support of this view, we showed that 10 µM of the CaMKII inhibitor KN93, acutely added to the Hypo solution, was able to reduce ICl,_swell_ by ~40% in MLC1-WT cells (Figure 5C,D). Notably, this was about the same difference between ICl,_swell_ recorded in MLC1-Ø and MLC1-WT or between T17D and T17A cells. Interestingly, in MLC1-Ø, as well as in cells expressing MLC1 proteins that cannot be modulated by the CaMKII-mediated phosphorylation (MLC1-T17D and MLC1-T17A), KN93 was not able to reduce the hyposmosis-induced ICl,_swell_.

### 3.6. Hyposmotic Challenge Induces Ca^2+^ Influx and Ca^2+^ Release in U251 Cells

It is known that CaMKII is activated by the Ca^2+^ binding protein Calmodulin upon increase of cytosolic Ca^2+^. Increased cytosolic Ca^2+^ due to cell swelling has also been shown to activate ICl,_swell_ in astrocytes [42,43]. These results identify Ca^2+^ as a key player in the activation pathways of both MLC1 and ICl,_swell_. Since specific Ca^2+^ sources are known to be linked to specific functions, we aimed to distinguish the roles of external Ca^2+^ influx and Ca^2+^ release from internal stores in the CaMKII/MLC1-dependent enhancement of the ICl,_swell_. First, we carried out Fura-2-based Ca^2+^ recording experiments, to discriminate between the contribution of external Ca^2+^ influx and intracellular Ca^2+^ release in the hyposmosis-induced Ca^2+^ movements in U251. A hyposmotic challenge lasting 15 min was imposed on Fura-2-loaded MLC1-WT cells in the presence or absence of external Ca^2+^ (Figure 6A–C). When control solution containing 2.5 mM Ca^2+^ was used, most cells responded with a fast and transient Ca^2+^ rise, very variable in amplitude. A slow rise of Ca^2+^ followed the initial peak, varying in both speed of onset and amplitude. On the other hand, in Ca^2+^-free solution, the initial peak was absent, and only a slowly raising phase, with a similar onset rate and amplitude, was observed. We can conclude that the first Ca^2+^ peak was mainly due to Ca^2+^ entry, with the possible minor participation of Ca^2+^ release from internal stores induced by Ca^2+^ entry (so called CICR), which, instead, was represented by the asynchronous Ca^2+^ transients predominant in the later phase. Finally, the slowly rising phase recorded in 0-Ca^2+^ depicts a homogeneous response among cells, due to release from unidentified intracellular stores, and is not attributable to a CICR mechanism. To offer a quantitative evaluation of the very variable response to hyposmotic challenge observed in the presence of Ca^2+^, we calculated the integral (i.e., area) of the Ca^2+^ signal at different time intervals, in both the presence and absence of extracellular Ca^2+^. The resulting graph (Figure 6D,E) confirms the initial Ca^2+^ raise, followed by a larger Ca^2+^ response, which decays at later times in the presence of Ca^2+^, and a slight and monophasic rise in the absence of Ca^2+^.

### 3.7. CaMKII/MLC1-Dependent Enhancement of ICl,_swell_ Requires Release of Ca^2+^ from Intracellular Stores Induced by Extracellular Ca^2+^ Influx

Once it was ascertained that the hypotonic stimulus triggered both Ca^2+^ influx and Ca^2+^ release from intracellular stores, we asked if changes in cytosolic Ca^2+^ levels were important for the CaMKII/MLC1-dependent enhancement of ICl,_swell_. We first carried out patch-clamp experiments aimed at distinguishing between the specific contributions of the two Ca^2+^ sources. To study the role of Ca^2+^ influx, after a stabilization period in 0-Ca^2+^ isotonic solution, cells were exposed to hyposmotic solution, in the absence of external Ca^2+^, to assess the amount of ICl,_swell_ activated by cell swelling without the contribution of Ca^2+^ influx. After a stable IICl,_swell_ activation was reached, cells were probed with a hypotonic solution containing 2 mM Ca^2+^, to evaluate a possible further increase of the current due to the entry of external Ca^2+^ (Figure 7A). The results obtained with the different cell models showed that only in cells expressing MLC1-WT was the ICl,_swell_ activated by the Ca^2+^-free hypotonic solution further increased (by ~35%) upon addition of external Ca^2+^, while in all the other cell types (MLC1-Ø, MLC1-T17D, and MLC1-T17A) the current density remained unchanged (Figure 7B and Appendix A). These results were confirmed by similar tests carried out with 0-Ca^2+^ and with 2 mM Ca^2+^ in the activating hyposmotic solution, but this time done in separate MLC1-WT cell groups. Moreover, in this case, we found that the hyposmosis-activated ICl,_swell_ density in the presence of extracellular Ca^2+^ (~54 pA/pF) was markedly greater than that elicited in the cell group probed in the absence of extracellular Ca^2+^ (~32 pA/pF) (Figure 7C). The above results suggest that additional activation of ICl,_swell_ by external Ca^2+^ influx occurred only where MLC1 activity could be modulated by Ca^2+^-dependent CaMKII, which was the major feature distinguishing MLC1-WT from all the other cell lines analyzed here. Overall, these experiments indicated that Ca^2+^ influx is needed for MLC1-dependent ICl,_swell_ potentiation in U251 cells. Then, we wanted to test whether Ca^2+^ release was also involved. To this end, we assessed the maximal ICl,_swell_ activated by hypotonic solution with 0-Ca^2+^ (to avoid contamination from external Ca^2+^ influx and ensuing activation of the CICR process) in MLC1-WT cells and in the same cells pretreated for 5 min with 1 µM Thapsigargin (TG), a non-competitive inhibitor of the ER Ca^2+^ ATPase (SERCA), known to cause a massive Ca^2+^ release from the ER (i.e., with internal stores fully depleted). A representative experiment carried out in WT cells pretreated with TG, with the hypotonic solution with 0-Ca^2+^ activated ICl,_swell_, is shown in Figure 7D. The same protocol was applied to WT cells not pretreated with TG, to be used as control. As shown in Figure 7E, in the absence of Ca^2+^ influx (0-Ca^2+^ condition) the TG pretreatment did not significantly alter the ICl,_swell_ density. As the influx of Ca^2+^ from extracellular space is able to further activate the ICl,_swell_ in WT cells, we then asked whether Ca^2+^ entering into the cell could directly enhance the ICl,_swell_ via the CaMKII/MLC1 pathway, or indirectly, by stimulating the release of Ca^2+^ from the intracellular stores via CICR. To this end, we assessed the ICl,_swell_ density activated by hypotonic solution with 0-Ca^2+^, in MLC1-WT cells pretreated or not with TG, and then verified if ICl,_swell_ density could be further increased upon switching to the hypotonic solution with 2 mM Ca^2+^ (Figure 7F; i.e., we followed the same protocol as the experiment shown in Figure 5). The analysis of these results revealed that only in cells not treated with TG, was the ICl,_swell_ activated by the Ca^2+^-free hypotonic solution further increased (by ~35%) upon addition of external Ca^2+^, while in TG-treated cells, the current density remained unchanged (Figure 7F). Similar results were obtained by including in the pipette solution the IP3 receptor (IP3R) inhibitor 2-APB (see Materials and Methods). These observations revealed that when the internal stores are fully depleted of Ca^2+^ or when the release of Ca^2+^ from IP3R is blocked, and no CICR process can be activated, the external Ca^2+^ influx per se is ineffective in increasing ICl,_swell_ Notably, Ca^2+^ influx potentiates ICl,_swell_ by activating the CICR mechanism only in cells expressing the CaMKII phosphorylatable MLC1 protein.

## 4. Discussion

Pathological astrocyte swelling often occurs in response to CNS damage, leading to cellular (cytotoxic) edema. Cytotoxic edema represents a first step in the formation of tissue edema, as it generates the driving force for the vasogenic edema formation, causing tissue swelling [7]. This pathological cascade may reflect MLC disease pathogenesis, where an increased water content in brain is due to mutations in MLC1, a protein favoring astrocyte regulatory volume decrease after hyposmosis-induced swelling [13,44]. Understanding MLC1 function and the molecular mechanisms regulating volume changes in astrocytes may pave the way for the identification of possible molecular target(s) to treat brain edema in MLC. Following this aim, in the present study we revealed that MLC1 functional activation, assessed by MLC1-mediated VRAC current potentiation, is favored by CAMKII phosphorylation of MLC1 in response to stimuli inducing intracellular Ca^2+^ increase in astrocytes. Since Ca^2+^ release is strongly linked to astrocyte activation in response to several pathological stimulations, these findings may have interesting implications for the comprehension of MLC1′s proper function and its central role in the control of volume changes and astrocyte activation processes in different brain pathological conditions, as discussed below. 

### 4.1. CamKII Binds and Phosphorylates MLC1

We, here, have provided evidence that CaMKII binds MLC1 at its COOH terminal, similarly to what has been observed for the Ca^2+^ channel TRPV4 [43]. In addition, IF staining showed that treatments inducing Ca^2+^ release in astrocytes (hyposmosis, high K^+^) favored MLC1/CaMKII interaction, suggesting a possible functional outcome of this interaction. The identification by proteomic analysis of the T17 aa as the CaMKII target site and the characterization of the U251 cell lines expressing the MLC1 mutant insensitive to CaMKII phosphorylation (T17A) or mimicking constitutive phosphorylation (T17D), proved that CaMKII favors MLC1 protein assembly (dimer formation), an indication of MLC1 functional activation [19,20,21,23,25,26], and increases protein half-life (stabilization), but does not alter MLC1 trafficking. The occurrence of phosphorylation in response to ATP stimulation, only in MLC1-WT, and not in MLC1-T17A mutant, confirmed that CaMKII also phosphorylates MLC1 in intact cells and not only in in vitro assays, in response to intracellular Ca^2+^ release. The functions of many proteins involved in volume regulation, such as AQP4 [39] and calcium (TRPV4) [45] and chloride channels (Best-1, Ano-1, VRAC, ClC3) [46,47,48], are modulated by CaMKII interaction and phosphorylation. Interestingly, the coexisting binding of the PP1 and PP2A phosphatases at the MLC1 C-terminal suggests that MLC1 protein may be subjected to dynamic phosphorylation/de-phosphorylation events, likely involved in the modulation of its functionality.

### 4.2. CaMKII-Mediated Phosphorylation of MLC1 Confers Ca^2+^-Dependence to VRAC Activation 

VRAC is the channel mainly responsible for both ICl,_swell_ and volume regulation in several types of cell. Once activated, VRAC is permeable to osmolytes such as taurine, glutamate, and Cl^–^ anions, whose efflux results in cell water outflow and restoration of the original cell volume [49,50]. Lack of functional MLC1 reduces VRAC currents [13,15]; conversely, MLC1 overexpression increases VRAC currents [51]. We here demonstrate that, in U251 cells, the overexpression of MLC1-WT, but not of the pathological mutant form (MLC1-S280L), affects several biophysical features of the current, such as its amplitude, inactivation degree, and kinetics. The biochemical and electrophysiological characterization of MLC1-T17A and T17D mutants, demonstrated that this specific post-translational modification is essential for MLC1 to modulate the biophysical and functional properties of VRAC and to confer Ca^2+^ dependence to this channel, which would be otherwise insensitive to Ca^2+^ ions in glioma cells [52]. The observation that, in presence of external Ca^2+^, ICl,_swell_ is increased in MLC1-WT, but not in T17A or T17D mutants, is in accordance with the ability of Ca^2+^ influx to promote CaMKII-dependent MLC1 phosphorylation only in MLC1-WT expressing cells. Moreover, the abolition of a Ca^2+^ influx-mediated increase of ICl,_swell_ after pretreatment with TG and 2-APB, confirmed the contribution of internal Ca^2+^ stores in this process, possibly via activation of the CICR mechanism. As summarized in Figure 8, our results are consistent with the following cascade of events: (1) induction of an initial Ca^2+^ influx by hypotonic cell swelling, most likely due to opening of stretch-activated Ca^2+^ channels (such as TRPV4); (2) triggering of Ca^2+^ release from intracellular stores through the CIRC mechanism; (3) Ca^2+^-induced CaMKII activation; (4) CaMKII-mediated MLC1 phosphorylation and consequent dimer stabilization of the newly activated protein and/or of that already present at the PM; (5) MLC1 potentiation of VRAC current. Besides the MLC1-mediated VRAC potentiation, the molecular mechanisms of VRAC activation is still an open question. The initial hypothesis involving changes in ion strength is now paralleled by other mechanisms, among which are membrane tension and signaling pathways. The heterogeneity of the activation mechanism may be attributed to cell-specific properties of VRAC and different interactions of the channel with Ca^2+^-sensitive effectors. In astrocytes, VRAC is also stimulated by neurotransmitters and neuromodulators such as ATP, through an increase of intracellular Ca^2+^, PKC, and CaM/CaMKII pathways activation [53]. Although these studies leave unresolved the mechanism by which PKC and CaM/CaMKII activate VRAC, they are in keeping with our data on U251 cells expressing MLC1 and the mechanism suggested above (Figure 8). We here propose MLC1 as the essential component of the Ca^2+^-dependent modulation of VRAC in astrocytes. This notion is further supported by the fact that in astrocytoma/glioblastoma cells, where MLC1 is virtually absent (or minimally expressed), changes in intracellular Ca^2+^ do not affect ICl,_swell_ [52]. On the contrary, in primary astrocytes where MLC1 is highly expressed, the Ca^2+^-modulation of the ICl,_swell_ was reported [42,43]. It would also be plausible to assume that other similar proteins may replace MLC1 in providing Ca^2+^-sensitivity to VRAC in other cell types where the Ca^2+^ dependence of ICl,_swell_ has been demonstrated. To date, no direct interaction between MLC1 and the LRRC8A protein, the most generally recognized molecular component of the VRAC channels [54,55], has been found in astrocytes and brain tissue [56]. In addition, we did not detect LRRC8A among the molecular interactors previously co-purified with MLC1 with the use of the Histidine-tagged protein affinity methods, or in the co-IP here presented (not shown), confirming the previously published evidence. However, MLC1 may act by indirectly influencing the post-translational modifications of some LRRC8 accessory subunits, forming the fully functional channel, in a cell specific way, as we previously observed for the LRRC8C subunits [56]. In light of the recent findings indicating a possible role for the Tweety-homologs proteins (Ttyh1-3) in the activation of the ICl,_swell_ in astrocytes [57], it will also be extremely interesting to study the relationship between MLC1 and this class of proteins in the future.

### 4.3. Significance of Astrocyte Ca^2+^ Signaling in MLC Pathogenesis and Beyond

Overall, the data here presented reveal that, in astrocytes, MLC1 protein function is regulated by Ca^2+^ changes occurring in response to osmotic stress. Interestingly, in different in vitro and in vivo models, osmotic stress has been found to elicit Ca^2+^ responses in astrocyte end-feet, where MLC1 is preferentially expressed [14,22]. We can hypothesize that the osmotic- or other stress signals inducing CICR in perivascular astrocyte end-feet lead to MLC1 activation and VRAC potentiation aimed at stimulating an efficient RVD. Alterations to these processes may cause abnormal astrocyte end-feet swelling, resulting in the dysfunction of both the blood–brain barrier (BBB) and of the neuro-vascular unit (NVU), the functional entity composed of neurons, astrocytes, pericytes, endothelial cells, and basal lamina that couples neuronal activity to cerebral blood flow in response to energy demand [58]. In addition, since adjacent astrocytes are interconnected via Ca^2+^-permeable gap-junctions, the CICR mechanism generates Ca^2+^ waves propagating through the astrocyte syncytium [59]. Ca^2+^ wave spreading is known to modulate several fundamental astrocyte activities (i.e., release/uptake of ions and gliotransmitters, support to BBB and NVU functionality) [58]. By potentiating VRAC channels, known to favor intercellular Ca^2+^ wave propagation via extracellular ATP release [60], and by favoring the function of the gap-junction protein Cx43 that connects adjacent astrocytes [61], MLC1 might be itself involved in the control of Ca^2+^ wave spreading. It is worth noting that alterations of Ca^2+^ diffusion at the perivascular end-feet can cause serious consequences for NVU and neuron functionality, since a Ca^2+^ increase in this compartment adjusts cerebral blood flow during synaptic activity [62]. Indeed, electrical field stimulation in brain slices and in vivo induces an increase in intracellular Ca^2+^ in perivascular astrocyte end-feet, followed by vascular smooth muscle Ca^2+^ oscillations and arteriolar dilation [63]. Thus, we cannot exclude that alterations of Ca^2+^ dynamics in astrocyte end-feet due to MLC1 mutations might influence arteriolar contractility and BBB permeability, causing edema and NVU dysfunctions. In accordance with this hypothesis, a recent study provided evidence of alterations of vascular smooth muscle contractility and NVU coupling in MLC1 KO mice [64]. The emerging role of MLC1 in the maintenance of BBB/NVU integrity has also been recently highlighted in an elegant in vivo study by Morales and collaborators [65], who generated and characterized transgenic mice where inducible and selective ablation of perivascular astrocytes expressing MLC1 (but not of other astrocyte populations) could be obtained.

Dysfunction of NVU and of cerebral blood flow are also implicated in the pathogenesis of cerebral disorders, such as traumatic brain injury, Alzheimer’s (AD) and Parkinson’s (PD) diseases, multiple sclerosis (MS), and glioblastoma malignancy [58,66]. Interestingly, increased expression of MLC1 at the perivascular astrocyte end-feet has been observed in the brain of patients affected by AD, MS, and also prion disease [31], which might indicate an attempt to ameliorate these pathological processes by increased MLC1 expression in activated astrocytes. In this scenario, it is noteworthy that Ca^2+^ release from intracellular stores occurs in astrocytes in response to different damaging stimuli, leading to astrocyte swelling and activation [67].

## 5. Conclusions

In the present study, we revealed for the first time that MLC1 is a Ca^2+^-sensitive protein linking VRAC-dependent volume regulation to Ca^2+^ dynamics in astrocytes. Although further analyses are needed to fully understand MLC1′s function and the details of MLC/VRAC cooperation, the results here presented open new perspectives for comprehension of the astrocyte-mediated molecular events governing brain water/ion exchanges in normal and pathological conditions. Considering the potential of astrocytes as pharmacological targets, this knowledge may be of interest for the identification of astrocyte-based therapeutic approaches for MLC and other brain diseases where astrocyte swelling and brain edema have a central role in the pathological process.

## Figures and Tables

**Figure 1 cells-11-02656-f001:**
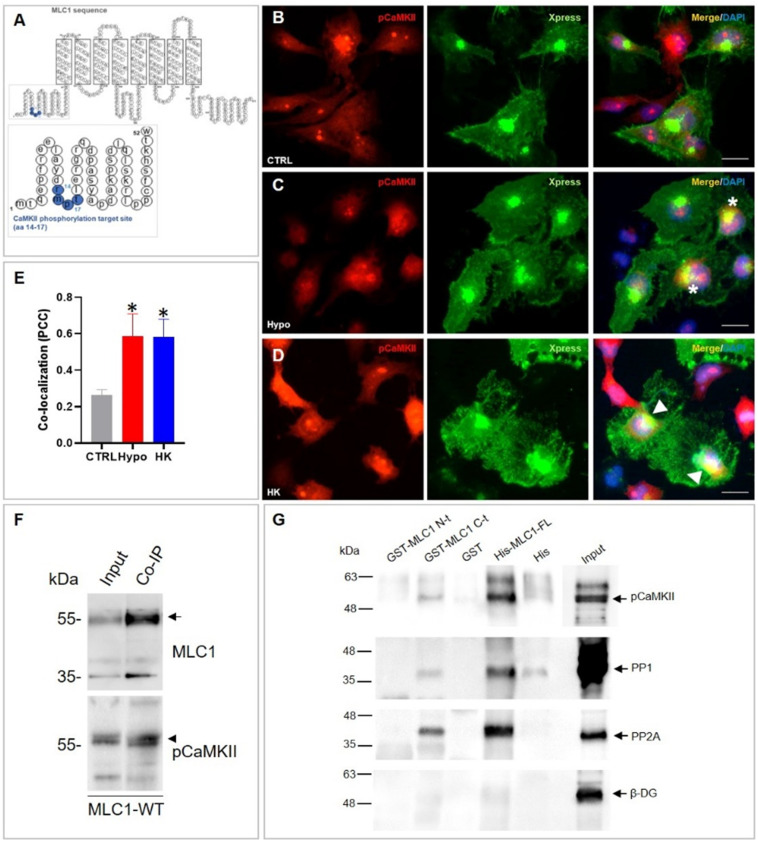
CaMKII binds the MLC1 protein in U251 cells. (**A**) Schematic representation of MLC1 protein sequence showing a potential CaMKII phosphorylation target site in the MLC1 NH_2_ terminal (aa 14–17). (**B**–**D**) IF staining of MLC1-WT U251 cells with anti-phospho (p)CaMKII pAb (red) in combination with anti-Xpress mAb (green) to detect the Xpress tag localized in the MLC1 NH_2_ terminal shows low levels of pCaMKII expression in the cytosol of U251 cells in control conditions (CTRL, (**B**)) and an increase of its expression and a partial colocalization with MLC1 in the perinuclear areas after a 30-min treatment with hyposmotic (Hypo) and high K^+^ (HK) solutions (asterisks and arrowheads in (**C**,**D**), respectively). Scale bars: 20 μm. Panel (**E**) shows the quantification of the colocalization signal (yellow) evaluated using ImageJ software. The degree of colocalization was estimated using the Pearson correlation coefficient (PCC). Data are shown as means ± SEM of 50–60 cells for each condition (* *p* < 0.001 vs. CTR, unpaired two-tailed Student’s *t*-test). (**F**) Immunoprecipitation (IP) of MLC1 protein from MLC1-WT U251 cells using the anti-Xpress mAb. Immunoblotting performed with the anti-MLC1 pAb (arrow) and anti-pCaMKII pAb (arrowhead) showed that MLC1/pCaMKII co-immunoprecipate. (**G**) A pull-down assay of rat brain extract was carried out with sepharose-immobilized GST-MLC1-NH_2_ and -COOH terminal domains (GST-MLC1-N-t and GST-MLC1-C-t, respectively), and with GST alone as control. Agarose-immobilized Histidine (His)-fused MLC1 full-length protein (His-MLC1-FL) and Histidines (His) alone, as a control, were also used. WB of protein eluate reveals the binding of pCaMKII, and of the phosphatases PP1 and PP2A to His-MLC1-FL and GST-MLC1-C-t, but not to GST-MLC1-N-t. Beta dystroglycan (β-DG) was used as negative control. The starting material derived from rat brain extract is indicated as Input. One representative experiment out of the three performed is shown.

**Figure 2 cells-11-02656-f002:**
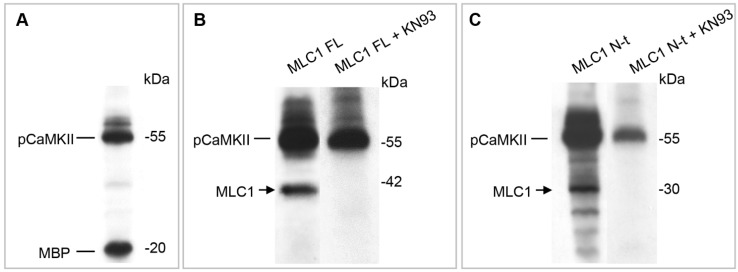
CaMKII phosphorylates threonine 17 residue in the MLC1 NH_2_ terminal domain. (**A**–**C**) In vitro kinase assay using CaMKII enzyme immunoprecipitated from U251 cells. The not-specific substrate MBP (**A**), the His-MLC1 full-length protein (FL, (**B**)), or the sepharose-immobilized GST-MLC1 NH_2_-terminal domain (MLC1 N-t, (**C**)) were incubated with the CaMKII enzyme, with or without the CaMKII inhibitor KN93, and the reaction was carried out in the presence of [γ^32^P] ATP. The [32P]-labeled proteins revealed by autoradiography indicated that MLC-FL and MLC1 N-t were specifically phosphorylated by CaMKII. (**D**) Mass spectrometry analysis of the phosphorylated MLC1 NH_2_ terminal peptide after GluC digestion. In the upper part of the figure, the sequence of GST-MLC1 NH_2_-terminal region is shown, with the MLC1-NH_2_-terminus highlighted in red. The lower panel shows the MS3 spectrum of the double-charged phosphorylated peptide LAYDRMPTLE obtained after phosphoric acid loss caused by CID fragmentation. This neutral loss confirmed the presence of a phosphorylated site. B and Y series ions (reported in red and blue, respectively) in the MS3 spectrum localize the phosphorylation at T17. Phospho-T (pT) in the peptide sequence indicates a phosphorylated T, while the asterisk marks the localization of the pT in the sequence of MLC1-N-terminal.

**Figure 3 cells-11-02656-f003:**
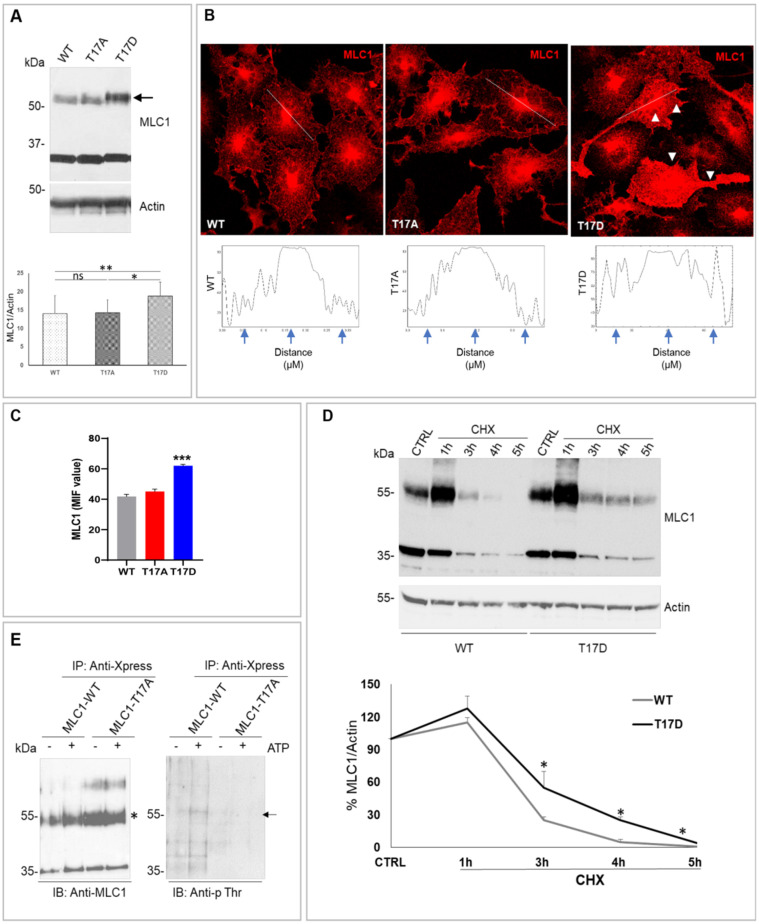
CaMKII-mediated phosphorylation of the MLC1 T17 residue favors MLC1 protein dimerization and stabilization. (**A**) WB analysis of U251 mutant cell lines shows that T17D substitution favors MLC1 dimer formation (arrow) when compared to MLC1-WT and T17A expressing cells. The bar graph below the WB represents the densitometry analysis of the MLC1 protein bands normalized with the amount of actin (means ± SEM of 3 independent experiments; * *p* < 0.05, ** *p* < 0.001 calculated using non-parametric test). Panel (**B**) shows IF staining of cells expressing MLC1-WT, T17A, and T17D mutants with anti-MLC1 pAb (red). An increase of the MLC1-T17D mutant localization at PM and in intracellular compartments (arrowheads in B) when compared to WT and T17A MLC1 is observed. Scale bars: 20 μm. Below each IF panel, the distribution of IF pixel intensity along a freely defined line (representatively indicated in each IF images) spanning the whole cell confirms a general increase of the MLC1-T17D protein fluorescence intensity. Fluorescence intensity peaks are marked by arrows. One representative intensity plot is shown for each IF panel. (**C**) Mean fluorescence intensity (MIF) of 50–60 cells/conditions from 3 independent experiments was calculated (means ± SEM values of mean, *** *p* < 0.0001 calculated using one-way ANOVA, unpaired two-tailed Student’s *t*-test). (**D**) WB analysis of U251 cells expressing MLC1-WT and the T17D mutant, untreated (CTRL) or treated with cycloheximide (CHX, 100 µg/mL) for 1, 3, 4, and 5 h revealed an increase of the MLC1-T17D protein half-life (stability) when compared to MLC1-WT. A graph indicating the densitometry analysis of MLC1 protein bands normalized with the amount of actin is shown below. Data are expressed as the percentage of the value measured in control untreated cells (means ± SEM of 3 replicates for each type of experiments; * *p* < 0.05 calculated using non-parametric test). (**E**) WB of protein eluates derived from immunoprecipitation of MLC1-WT and MLC1-T17A expressing cells with anti-Xpress mAb in control conditions or after ATP stimulation (5 min, 100 µM). Immunoblotting was performed with the anti-MLC1 pAb (asterisk), as positive control of IP procedures, and with anti-phosphothreonine (anti-p Thr) mAb to assess Thr phosphorylation levels. As indicated by the arrow, Thr phosphorylation signal increases after ATP stimulation in MLC1-WT, corresponding with the MLC1 protein molecular weight, and not in MLC1-T17A expressing cells.

**Figure 4 cells-11-02656-f004:**
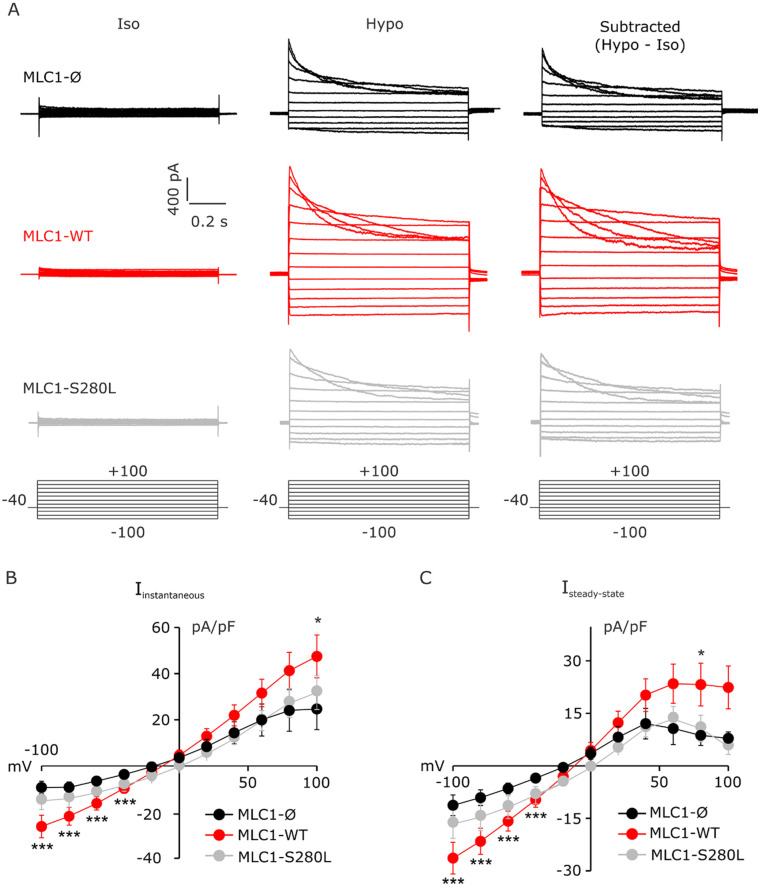
MLC1 upregulates ICl,_swell_ currents in U251 astrocytoma cells. (**A**) Representative I–V relationships recorded from −100 to +100 mV pulses (Delta 20 mV, holding potential −40 mV, duration 1 s) in U251 cells infected with an empty vector (Ø) (**top**), expressing MLC1-WT (**middle**) or MLC1-S280L mutant (**bottom**) in control condition (**left**), following application of hyposmotic solution (**center**) and after digital subtraction (Hypo-Iso, **right**). (**B**,**C**) Mean I–V relationships of instantaneous and steady state current obtained in MLC1-Ø (black; *n* = 6), MLC1-WT (red; *n* = 9), and MLC1-S280L cells (grey; *n* = 6) after digital subtraction (Hypo-Iso) (mean ± SEM; * *p* < 0.05; *** *p* < 0.001).

**Figure 5 cells-11-02656-f005:**
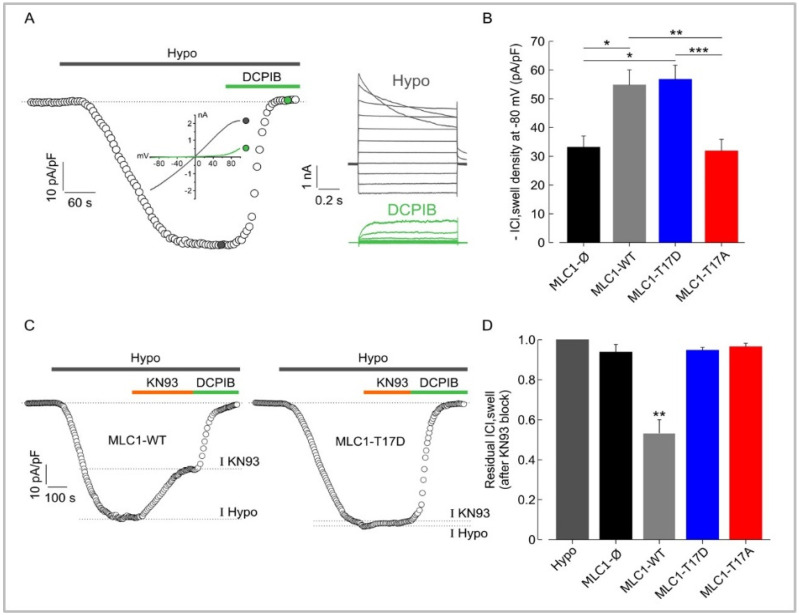
Phosphorylation of MLC1-T17 potentiates ICl,_swell_ in U251 cells. (**A**) Representative time course of ICl, swell density (pA/pF) measured at −80 mV (the equilibrium potential for K+ under our recording conditions), taken from current ramps (from −100 to 100 mV, 600 ms duration, holding potential −40 mV) applied during the application of hypotonic solution (grey trace) and upon addition of 10 µM DCPIB (green trace). Right: families of current traces evoked by applying 1 s voltage steps from −100 to 100 mV, in steps of 20 mV (HP −40 mV) in the presence of a hypotonic solution (grey traces), and upon addition of 10 µM DCPIB (green traces). (**B**) Bar plot showing the average current density measured at −80 mV during exposure to 30% hypotonic solution, in U251 cells transfected with the empty vector (MLC1-Ø, *n* = 10), with MLC1-WT protein (*n* = 14), and with MLC1-T17D (phosphorylation mimicking mutant, *n* = 9) or MLC1-T17A (not phosphorylatable mutant, *n* = 14). (**C**) Representative time courses of ICl,_swell_ density (pA/pF) measured at −80 mV from current ramps, during application of hypotonic solution (grey bar) and upon addition of 10 µM KN93 and 10 µM DCPIB, in MLC1-WT (**left**) and MLC1-T17D (**right**) U251 cells. The dashed lines in the time courses indicate the levels where the peaks of currents were measured. (**D**) Bar plot showing the mean residual ICl,_swell_ in the presence of 10 µM KN93, as normalized to the ICl,_swell_ elicited during exposure to 30% of hypotonic solution in the absence of the KN93 (Hypo), in the different cell lines (Mean ± SEM *n* = 3–5; * *p* < 0.05; ** *p* < 0.01, *** *p* < 0.001).

**Figure 6 cells-11-02656-f006:**
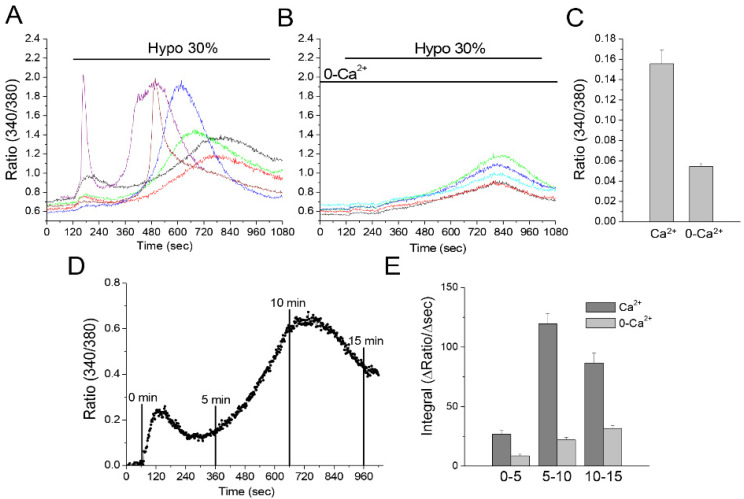
Ca^2+^ movements in U251 cells in response to hyposmotic challenge. Fura-2-loaded MLC1-WT expressing U251 cells were exposed to hyposmotic solution for 15 min in the presence of Ca^2+^ (2.5 mM), or in a 0-Ca^2+^ solution (Ca^2+^ replaced by Mg^2+^ and with EGTA 0.5 mM). Exemplificative traces recorded in the presence (**A**) or absence of Ca^2+^ (**B**) are shown. The amplitude of the Ca^2+^ peak was calculated as the difference between the maximum value reached within the first 120 s of application of hyposmotic solution and the baseline value. Peak amplitudes are plotted in panel (**C**) (mean ± SEM of 90 and 65 cells, from 3 recordings for each condition). Areas underneath Ca^2+^ transients at defined time intervals (0–5, 5–10 and 10–15 min) were calculated after baseline subtraction, as shown in panel (**D**), to represent actual Ca^2+^ movements. Mean ± SEM of ΔRatio/Δtime at different time intervals are shown in panel (**E**) (*n* = 72 and 65 cells, in Ca^2+^ and 0-Ca^2+^, respectively).

**Figure 7 cells-11-02656-f007:**
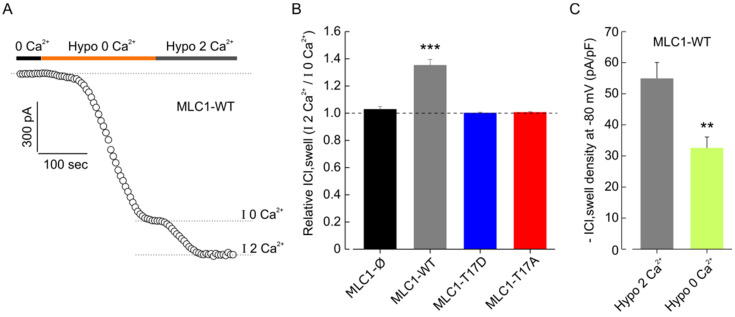
Extracellular and endoplasmic reticulum Ca^2+^-dependence of ICl,_swell_ in U251 cell lines carrying WT and mutated MLC1. (**A**) Representative time course of ICl,_swell_ during application of a Ca^2+^-free hypotonic solution (Hypo 0 Ca^2+^) and a hypotonic solution containing 2 mM Ca^2+^ (Hypo 2 Ca^2+^). The dashed lines indicate the levels where the maximal hypotonic-activated current in the absence (I 0 Ca^2+^) and presence (I 2 Ca^2+^) of extracellular Ca^2+^ were measured. (**B**) Bar plot showing the mean ICl,_swell_ current elicited by the addition of 2 mM external Ca^2+^ (dashed lines), normalized to the current activated by Hypo 0 Ca^2+^, in MLC1-Ø (*n* = 5), MLC1-WT (*n* = 8), MLC1-T17D (*n* = 6), and MLC1-T17A (*n* = 5) U251 cell lines (mean ± SEM, *** *p* < 0.01, using one-sample *t*-test). (**C**) Bar plot showing the average current density during exposure to hypotonic solution, in the continuous presence (grey bar, *n* = 14) or absence (light green bar, *n* = 9) of external Ca^2+^, assessed in MLC1-WT cells (mean ± SEM, ** *p* < 0.01, using two-sample *t*-test). (**D**) Representative time course of ICl,_swell_ during application of a Ca^2+^-free hypotonic solution (Hypo 0 Ca^2+^) followed by a hypotonic solution containing 2 mM Ca2+ (Hypo 2 Ca^2+^), after 1 μM of thapsigargin (TG) pretreatment (5 min to deplete ER Ca^2+^stores). (**E**) Bar plot showing the average current density during exposure to hypotonic 0 Ca^2+^ solution, in MLC1-WT cells in the absence (CTRL, *n* = 9) and presence (TG, *n* = 9) of TG pre-treatment (mean ± SEM, *p* = 0.36). (**F**) Bar plot showing the mean fractional ICl,_swell_ elicited by Hypo 2 Ca^2+^, normalized to the current activated by Hypo 0 Ca^2+^, in MLC1-WT cells either in the absence (*n* = 14) or presence (*n* = 9) of TG or of the IP3R inhibitor 2-APB (*n* = 7) (mean± SEM, ** *p* < 0.01).

**Figure 8 cells-11-02656-f008:**
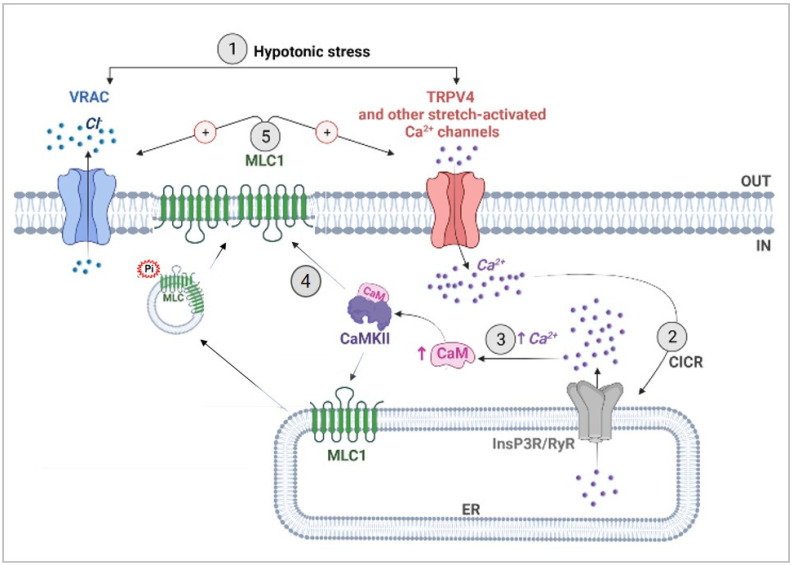
Scheme showing the proposed molecular mechanisms of the MLC1-induced Ca^2+^ regulation of VRAC. (1) Exposure to a hypotonic stress induces astrocyte (or astrocyte end-feet) swelling, which triggers the opening of TRPV4 and other Ca^2+^ channels (voltage-dependent and mechanosensitive Ca^2+^ channels). (2) The entry of extracellular Ca^2+^ promotes Ca^2+^ release from the ER, likely mediated by the InsP3 (or/and Ryanodine) receptors, through the CICR mechanism. (3) The consequent elevation of intracellular Ca^2+^ is sensed by the Ca^2+^-binding protein calmodulin (CaM), which binds and activates CaMKII. (4) By binding MLC1 and phosphorylating the threonine aa at position 17 (T17), CaMKII promotes MLC1 dimerization, stabilization, and functional activation. (5) Phosphorylated MLC1 potentiates TRPV4 [20] and VRAC channels, favoring ICl,_swell_ and regulatory volume decrease.

## Data Availability

The data that support the findings of this study are available upon request from the authors.

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
