# Peer review of "The CaMKII/MLC1 Axis Confers Ca2+-Dependence to Volume-Regulated Anion Channels (VRAC) in Astrocytes"

_cells, 2022, doi:10.3390/cells11172656_

Round 1

Reviewer 1 Report

In the present work, the authors have combined molecular and biochemical techniques with electrophysiology and imaging to show that MLC1—a protein highly expressed in perivascular astrocytes—is a target of the Ca2+/Calmodulin–dependent protein kinase II. Further experiments have shown that the MLC1 phosphorylation occurring as a response to intracellular Ca2+ release potentiates VRAC–mediated ICl,swell. Finally, the authors have concluded that that MLC1 is a Ca2+-regulated protein linking volume regulation to Ca2+ signalling in astrocytes. The study is interesting and overall sound, the western blot results are well presented and the findings are novel. There are, however, some concerns that need to be addressed by the authors. 

1. The isotonic solution used by the authors in the control experiments is bound to have an osmolarity of approximately 330 mOsmol/l, which is higher than the regular cell culture media. Mind you, the addition of 30% distilled water for the preparation of a hypotonic solution significantly changes the concentrations of ions and other compounds. Did the authors measure the osmolarity of the used media and solutions? The authors must state this in M&M and also briefly re-iterate in M&M the specific maintenance protocol of the cell culture media/solutions regardless of the cited references.

2. Line 21 of the abstract: The abbreviation MC1 is unclear.

3. Different holding potentials can affect the current profile—the experiments have not been conducted with protocols of the same holding potential (–30 or –40mV?).

4. Supplementary information Figure SD1: The textual explanation differs from the protocol represented above (was the holding potential –40 or –30mV?)

5. Suggestion for Figure 6: In order to catch not only the differences in the initial calcium response but also in the late one, the authors may also calculate the integrated calcium (over the time after introduction of the hypotonic solution). The authors may show at least three intervals: 0—5, 5—10, and 10—15 min post adding the hypotonic solution.

6. Figure 6: Has the baseline fluorescence ratio between 0.6—0.7 been performed in all three experiments? The authors may consider estimating the ratio deltaF/F0 and then comparing this ratio for all three experiments (in Figure 6, the authors have presented quantification from one coverslip imaged)

7. Figure 7 A and D: The traces should be set at the same scales for a better visual comparison.

8. A reference to the supplementary data seems to be missing in the main text.

9. Regarding astrocytes, the release of calcium from the ER goes mainly, if not exclusively, via IP3R. The manuscript might benefit tremendously from an additional experiment that narrows down the receptor at the ER, e.g. using the same protocol as in the experiment with thapsigargin (Fig 7D), but with an IP3R blocker instead.

Author Response

Point-to-point replay letter

We thank both Reviewers for their insightful and helpful suggestions/comments. Below are reported answers (written in red) made by the authors to their criticisms.

Reviewer #1

In the present work, the authors have combined molecular and biochemical techniques with electrophysiology and imaging to show that MLC1—a protein highly expressed in perivascular astrocytes—is a target of the Ca2+/Calmodulin–dependent protein kinase II. Further experiments have shown that the MLC1 phosphorylation occurring as a response to intracellular Ca2+ release potentiates VRAC–mediated ICl,swell. Finally, the authors have concluded that MLC1 is a Ca2+-regulated protein linking volume regulation to Ca2+ signaling in astrocytes. The study is interesting and overall sound, the western blot results are well presented and the findings are novel. There are, however, some concerns that need to be addressed by the authors. 

  1. The isotonic solution used by the authors in the control experiments is bound to have an osmolarity of approximately 330 mOsmol/l, which is higher than the regular cell culture media. Mind you, the addition of 30% distilled water for the preparation of a hypotonic solution significantly changes the concentrations of ions and other compounds. Did the authors measure the osmolarity of the used media and solutions? The authors must state this in M&M and also briefly re-iterate in M&M the specific maintenance protocol of the cell culture media/solutions regardless of the cited references.

The actual osmolarity of the DMEM culture medium that we used for cell cultures is guaranteed by the factory in the range of 317-351 mOsm, which is not significantly different from the extracellular solutions used in our experiments. For instance, the calculated osmolarity of the extracellular solution used in our electrophysiological experiments was 317 mOsm which perfectly falls in the above-stated range. Unfortunately, we do not have the possibility of measuring the real osmolarity of our solutions. In addition, we are well aware that in our hypotonic solution we are significantly changing the ionic composition of the medium. However, we would like to underline that the method of inducing cell swelling by the addition of 30% distilled water is a well-established system and widely used by us and others in several previous published papers (Catacuzzeno et al., 2014; Sforna et al., 2017 and 2021; Centeio et al., 2020; Stuhlman et al., 2018).

As the Reviewer suggested we have now expanded the Materials and Methods section, by reporting the details of the recording solution.

  1. Line 21 of the abstract: The abbreviation MC1 is unclear.

We apologize for this inattention. We have now fixed this typo.

  1. Different holding potentials can affect the current profile—the experiments have not been conducted with protocols of the same holding potential (–30 or –40mV?).

We apologize for this mistake. Actually all the experiments (those in Fig. 7 as well as those in Figs. 8 and 9) were done using an holding potential of -40 mV. There was a typo in the legend of Figure 7. We have now fixed this typo as the Reviewer can appreciate.

  1. Supplementary information Figure SD1: The textual explanation differs from the protocol represented above (was the holding potential –40 or –30mV?)

See the answer for comment 3.

  1. Suggestion for Figure 6: In order to catch not only the differences in the initial calcium response but also in the late one, the authors may also calculate the integrated calcium (over the time after the introduction of the hypotonic solution). The authors may show at least three intervals: 0—5, 5—10, and 10—15 min post adding the hypotonic solution.

As suggested by the reviewer, we calculated the integral of the Ca2+ signal over time at the time intervals suggested (0-5, 5-10, 10-15 min), and for both Ca2+ containing and Ca2+ free solutions. An exemplifying trace and a bar graph are included as fig. 6D,E. A description of the analysis and its interpretation is now included in the manuscript (lines 499-504).

  1. Figure 6: Has the baseline fluorescence ratio between 0.6—0.7 been performed in all three experiments? The authors may consider estimating the ratio delta F/F0 and then comparing this ratio for all three experiments (in Figure 6, the authors have presented quantification from one coverslip imaged)

As pointed out by the reviewer, the baseline (i.e. before hypo-osmotic challenge) values of Ratio (340/380) are around 0.6-0.7 with some variability. In all the experiments, some min after positioning the coverslip in the recording chamber we used to start the recording of Fura-2 fluorescence with at least 60 sec in the control condition, to obtain the so-called baseline condition, which is used for calculating Ca2+ changes in offline analysis. So we do in all our recordings. In the previous version, we analyzed only one exemplificative experiment of the three performed. Now all the experiments have been analyzed and the results are included in the bar graph in fig. 6C.

  1. Figure 7 A and D: The traces should be set at the same scales for a better visual comparison.

We agree with the Reviewer observation in showing the two-time courses with the same scales. We have now updated Figure 7 by providing a rescaled trace in panel D, which now matches that of panel A.

  1. A reference to the supplementary data seems to be missing in the main text.

All the references in the text have been checked.

  1. Regarding astrocytes, the release of calcium from the ER goes mainly, if not exclusively, via IP3R. The manuscript might benefit tremendously from an additional experiment that narrows down the receptor at the ER, e.g. using the same protocol as in the experiment with thapsigargin (Fig 7D), but with an IP3R blocker instead.

We want to thank the Reviewer for raising this important issue. We have now performed experiments to verify the involvement of the IP3R in the modulation of VRAC through the MLC1-induced Ca2+-induced Ca2+ release (CICR). We first tried to perform experiments of extracellular Ca2+  re-addition in the constant presence of 100 µM 2-APB (which is the concentration recognized to fully inhibit IP3R) by adding the compound to the extracellular solution. Unfortunately, the addition of 2-APB to the extracellular solution directly inhibits VRAC current in U251 glioblastoma cells, in line with that reported previously by Lemonnier et al., 2004. We thus decided to perform additional experiments in which 2-APB was included in the pipette solution filling the patch electrode at a concentration 10 times higher than that needed to block IP3R (i.e. 1 mM). The rationale of this approach was to obtain a sufficiently high 2-APB concentration in the cytoplasm where IP3R is expressed. With this approach, VRAC current was not blocked anymore by 2-APB, and the upregulation induced by the extracellular Ca2+ re-addition was prevented, strongly suggesting that the CICR mechanism is mediated by IP3R receptor.

We have now reported these new data in the text and in the Figure 7F of the revised manuscript.

Reviewer 2 Report

Maria et al. have demonstrated that MLC1 regulates the astrocytic VRAC current through the CaMKII-mediated phosphorylation upon the increase of intracellular Ca2+ level. These days, the molecular identity and mode of action of astrocytic VRAC have been under debate. In this regard, the findings seem timely and novel, and the study is quite straightforward and well organized. However, there are major concerns to be addressed and/or revised before publication in Cells.

  1. All of the experiments were conducted in the U251 cell line, which does not represent physiological astrocyte biology. The authors should repeat the key experiments in primary cultured astrocytes (e.g., CaMKII-mediated phosphorylation of MLC1 and MLC1-mediated ICl,swell current with KN93, DCPIB, and different Ca2+ concentration.)

  2. Is MLC1 endogenously expressed in the U251 cell line? If so, studying the impact of gene-silencing or pharmacological blockade of MLC1 on ICl,swell current should be necessary.

  3. Please discuss the current debate on the identity of astrocytic VRAC: LRRC8A (Yang et al., PMID 30982627) and Ttyh (Han et al., PMID 31138989; Okada, PMID 31138989).

  4. It could be very interesting and helpful for the researchers in this field if the authors do some more experiments on the interaction between MLC1 and Lrrc8a/Ttyh.

  5. In the abstract, MC1 seems to be a typo. It could be corrected to MLC1.

Author Response

Point-to-point replay letter

We thank both Reviewers for their insightful and helpful suggestions/comments. Below are reported answers (written in red) made by the authors to their criticisms.

Reviewer #2

Maria et al. have demonstrated that MLC1 regulates the astrocytic VRAC current through the CaMKII-mediated phosphorylation upon the increase of intracellular Ca2+ level. These days, the molecular identity and mode of action of astrocytic VRAC have been under debate. In this regard, the findings seem timely and novel, and the study is quite straightforward and well organized. However, there are major concerns to be addressed and/or revised before publication in Cells.

  1. All of the experiments were conducted in the U251 cell line, which does not represent physiological astrocyte biology. The authors should repeat the key experiments in primary cultured astrocytes (e.g., CaMKII-mediated phosphorylation of MLC1 and MLC1-mediated ICl,swell current with KN93, DCPIB, and different Ca2+concentration).

We agree with the reviewer that confirmation of these experiments in primary astrocytes is an important issue.

For this reason, to demonstrate the functional interaction between MLC1 and the CaMKII in primary cells, we performed co-immunoprecipitation experiments using primary mouse astrocyte protein extracts and the pCaMKII antibody. These experiments revealed that also in these cells the activated, phosphorylated CaMKII interacts and co-immunoprecipitates with the MLC1 protein, leading to an enrichment of this latter in protein eluates. These results are shown in supplementary Figure 1 and described in the text on lines 259-262 of the revised version of the manuscript.

In addition, to further confirm the effect of pCaMKII on the stabilization of the recombinant (U251) and endogenous (primary astrocytes) MLC1 dimeric protein, we treated both cell types with the CaMKII inhibitor KN93 in presence of the hyposmotic stimulation known to favor MLC1 dimer formation. These experiments showed that CaMKII inhibition affects the formation/stabilization of the dimeric/plasma membrane-associated form of the MLC1 protein in both cell types, further confirming that the functional CaMKII has the same effects on endogenous and recombinant MLC1 protein (results are shown in supplementary Fig. 2 and described in the text at lines 358-361).

Unfortunately, the identification of MLC1 protein phosphorylation in primary cultures is very challenging due to the lower levels of MLC1 expression when compared to those obtained with the in vitro systems. However, the specificity of the CaMKII target site found at the MLC1 Nh2 terminal and the new results obtained on primary mouse astrocytes and above described, strongly support the evidence of MLC1 protein interaction with the activated form of the CaMKII and its consequent phosphorylation.

As suggested by the reviewer, it would be very interesting to study the effects of KN93, DCPIB, and different Ca2+ concentrations on MLC1-mediated ICl,swell current in primary astrocytes. However, to specifically and efficiently evaluate the contribution of the MLC1 protein, these experiments would require the use of a KO systems like astrocyte from KO mice that are not available at the moment in our laboratory.

However, we have recently developed iPSC-derived human astrocyte lines from 4 patients and 4 healthy control that, as suggested by the Reviewer, will be used in the next months to characterize the Ca2+ dependence of VRAC currents in a human primary cell system in a context of a new manuscript.

  1. Is MLC1 endogenously expressed in the U251 cell line? If so, studying the impact of gene-silencing or pharmacological blockade of MLC1 on ICl,swell current should be necessary.

In the U251 cells, the endogenous MLC1 is undetectable when compared to MLC1 transfected cells (Lanciotti et al. 2012).

  1. Please discuss the current debate on the identity of astrocytic VRAC: LRRC8A (Yang et al., PMID 30982627) and Ttyh (Han et al., PMID 31138989; Okada, PMID 31138989).

We thank the reviewer for this suggestion. We have commented on this new evidence in the Discussion section (673-676).

  1. It could be very interesting and helpful for the researchers in this field if the authors do some more experiments on the interaction between MLC1 and Lrrc8a/Ttyh.

As indicated in the Discussion section (lines 665-676), in our previous experiments aimed at isolating MLC1 interactors from U251 cells and primary mouse astrocytes by co-purification, immunoprecipitations, and pull-down (including those shown in the present paper), followed by western blot and, in some case, by proteomic (MALDI) analysis, we never found LRR8 protein subunits among MLC1 molecular partners. Our hypothesis is that MLC1 can modulate VRAC functions by influencing the post-translation modification of some LRR8 subunits (as we reported in Elorza-Vidal et al., 2018).

We thank the reviewer for the significant suggestion about the possible involvement of the Ttyh protein family in the MLC1-mediated regulation of ICl,swell.. As we discussed in the Discussion section (Lines 673-676), it will be very interesting to study in the next future the relationship between MLC1 and these class of proteins.

  1. In the abstract, MC1 seems to be a typo. It could be corrected to MLC1.

The typo has been corrected.

Round 2

Reviewer 1 Report

Great job!

Author Response

We thank the reviewer very much for the positive comments.

Reviewer 2 Report

This reviewer appreciates the authors' effort on revising the manuscript by adding some important findings. I recommend accepting this manuscript as is.

Author Response

(The authors gave the same response as above.)
